

# Axion-like particle as cold dark matter via the misalignment mechanism with PQ symmetry unbroken during inflation

Pawel Kozow⋆ and Marek Olechowski

Institute of Theoretical Physics, Faculty of Physics, University of Warsaw, Warsaw, Poland

⋆ pkozow@fuw.edu.pl

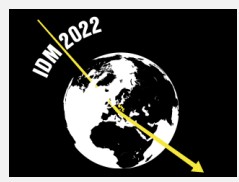

## Abstract

The QCD axion and axion-like particles (ALPs) are well motivated candidates for Cold Dark Matter (CDM). Such models may be divided into two classes depending on whether the associated Peccei-Quinn (PQ) symmetry is broken or not during inflation. The latter case is usually considered to be quite simple with relic density depending only on the corresponding decay constant and with no constraints from the known bounds on isocurvature perturbations. We will show that the situation is much more complicated. We will discuss conditions which should be fulfilled by ALP models with U(1) unbroken during inflation to be phenomenologically interesting.

## 1 Introduction

The PQ mechanism (see a review on axion cosmology, e.g. [1]) relies on an introduction of a global chiral $U(1)_{PQ}$ symmetry, under which a charged complex scalar $\Phi_{PQ}$ transforms. The $U(1)_{PQ}$ is spontaneously broken. Usually the following simple potential for $\Phi_{PQ}$ is considered

$$V_{PQ} = \lambda \left( \Phi_{PQ}^{\dagger} \Phi_{PQ} - \frac{1}{2} f_a^2 \right)^2 , \tag{1}$$

where $f_a$ denotes the spontaneous symmetry breaking scale and $a$ will denote the corresponding (pseudo) Goldstone boson, the QCD Axion; $\Phi_{PQ} = (1/\sqrt{2})S\, e^{ia/f_a}$. The QCD axion mass is generated later by non-perturbative potential $V_a$ which reads

$$V_a = \Lambda_{\mathrm{QCD}}^3 m_u \left( 1 - \cos \frac{a}{f_a} \right) \equiv m_a^2 f_a^2 (1 - \cos \theta) , \tag{2}$$

where $\Lambda_{\text{QCD}} \approx 0.2$ GeV, $m_u$ is the $u$-quark mass, which implies $m_a \simeq 6 \cdot \left(10^6 \text{ GeV}/f_a\right)$ eV. Astrophysical data constrain $f_a$ to roughly $10^9$ GeV $\lesssim f_a \lesssim 10^{17}$ GeV. Generalization to an ALP $\varphi$ results in considering its mass $m_\varphi$ as an additional free parameter (of some underlying non-perturbative sector characterized by scale $\Lambda_\varphi$). In what fallows we will simply use "the axion" to cover both the QCD axion and an ALP and $V_a$ to denote the corresponding potential.

The crucial assumption of the misalignment mechanism is that initially the axion is displaced from the minimum of its potential (2), i.e. $\theta_i \neq 0$. The onset of the corresponding oscillation of $a$ happens when the Hubble parameter decreases to $H \simeq (1/3)m_a$. The amplitude of oscillations soon starts to decrease with the scale factor as $\sim R^{-3/2}$ and at the particle level the oscillating field $a$ can be interpreted as a condensate of cold $a$-particles. Their interactions with thermal bath are suppressed by powers of $1/f_a$. Axion condensate constitutes a CDM candidate. For the QCD axion the onset of oscillations happens around $T = 1$ GeV. The CDM relic density is saturated by the QCD axion for $f_a \simeq 10^{11}$ GeV with $\theta_i \simeq 1$.

The $\theta_i$ depends on the inflationary dynamics of $\Phi_{PQ}$. There are two distinct cases which are specified by the hierarchy between $f_a$ and the Hubble parameter during inflation $H_I$. In the so-called "PQ broken scenario" one requires $f_a \gg H_I$. Here $|\Phi_{PQ}| = f_a$ during inflation [2]; the $\theta_i$ is fixed in our Universe, but random. The axion direction is massless during inflation. In turn, quantum fluctuations of each mode $k$ that leaves the horizon ($k \lesssim R H_I$) "freeze". Hence, during inflation stochastic fluctuations $\delta\theta$ to the homogeneous component $\theta_i$ are generated over Hubble volumes. The average squared grows linearly with e-folds $\Delta N_e$, $\langle\delta\theta^2\rangle = H_I^2/4\pi^2 f_a^2 \cdot \Delta N_e$. The larger $f_a$, the slower $\delta\theta$ dynamics. The stochastic fluctuations $\delta\theta$ become CDM relic density isocurvature fluctuations after $V_a$ is generated, later in the post-inflationary Universe. There are strong constraints on the CDM isocurvature power. For the QCD axion CDM ($\theta_i \simeq 1, f_a \simeq 10^{11}$GeV) these imply $H_I \lesssim 10^7$ GeV while for the ALP CDM one finds $H_I \lesssim 10^{10}$GeV. Notice, that the upper bounds on $H_I$ are much stronger than $O(10^{14})$ GeV obtained from constraints on tensor perturbations in the CMB.

Therefore of interest is the complementary so-called "PQ unbroken scenario" in which one requires $f_a \ll H_I$. The traditional approach [1] assumes that the thermal mass correction $(\alpha/24)T_{GH}^2\Phi_{PQ}^\dagger\Phi_{PQ}$, due to the Gibbons-Hawking temperature $T_{GH} = H_I/2\pi$, restores $U(1)_{PQ}$ during inflation and $\Phi_{PQ}$ attains zero. It is often assumed that no stochastic fluctuations are generated. The $U(1)_{PQ}$ is broken later, after inflation, when $T$ drops below the critical temperature. After the corresponding $U(1)_{PQ}$ phase transition, large spacial fluctuations are generated $\langle\delta\theta^2\rangle = \pi^2/3$. The fluctuations have "white noise" power [3]. The corresponding axion CDM isocurvature appears on relatively small scales. In particular, for the QCD axion the so-called "classical axion window" for misalignment, i.e. $f_a \simeq 10^{11}$ GeV and $10^{12}$ GeV $\lesssim H_I \lesssim 10^{14}$ GeV, is unconstrained by such isocurvature.

We would like to stress that the traditional approach to the "PQ unbroken scenario" is over-simplified and often leads to incorrect conclusions. Indeed, the thermal mass of the radial mode $V_{PQ}''(0) = \alpha H_I^2/96\pi^2 \ll H_I^2$ [4]. As a consequence, $\Phi_{PQ}$ features the stochastic dynamics during inflation, $\delta\Phi_{PQ}$ emerges. Inflationary evolution of light fields can be studied using the so-called "stochastic approach" [5]. Importantly, evolution of a probability distribution $P(t,\chi)$ of finding a light field $\chi$, coarse-grained over $H_I$ volumes, in $d^n\chi$ (which equals $P(t,\chi)d^n\chi$), can be derived. $P(t,\chi)$ satisfies the Fokker-Planck equation. The latter has a stationary solution, $P(\chi) \propto \exp\left\{-\frac{8\pi^2}{3H_I^4}V(\chi)\right\}$. For example, for $V_{PQ}$ described by eq. (1), the $P(t,\Phi_{PQ})$ approaches $P(\Phi_{PQ})$ after $\Delta N_e \simeq 15. \times \left(\frac{1}{\lambda}\right)^{1/2}$ e-folds [2]. For $\lambda \gtrsim 0.05$ this happens within $\Delta N_e < 60$. In turn, since $P(\Phi_{PQ})$ respects $U(1)_{PQ}$, one obtains that $\langle\delta\theta^2\rangle \simeq \pi^2/3$. Even if $U(1)_{PQ}$ is not thermally restored after inflation, the traditional over-simplified approach to the "PQ unbroken scenario" could lead to correct results if $\lambda \gtrsim 0.05$, but discrepancies are expected for smaller $\lambda$. For the case $\lambda \ll O(0.01)$ it is customary to assume long enough inflation

so that $P(\Phi_{PQ})$ is achieved *before* the last 60 e-folds of inflation over Hubble volumes of order of our Universe when it crosses the horizon. Hence, the typical initial (homogeneous) value of the radial mode, $S_i$, corresponds to the region around the maximum of $P(S) \propto S \exp\left(-\frac{8\pi^2}{3H_I^4} V(S)\right)$. More explicitly, one finds $S_i \equiv \langle S \rangle \simeq 0.3 \cdot H_I/\lambda^{1/4}$ for the quartic potential.

Sometimes very large $S_i$ is found useful. Correspondingly, very small $\lambda \lesssim 10^{-20}$ are considered, for example, in the context of Kinetic Misalignment [6] or Parametric Resonance [7] Mechanisms of axion DM production. However, if the coupling is so small, radiative corrections (from other couplings) could become relevant. The goal for the remaining of this work is to examine the role of radiative corrections to $\Phi_{PQ}$ (post)inflationary dynamics.

## 2 The radiative potential for $\Phi_{PQ}$ during inflation

In order to examine the role of radiative corrections, we will take the Gildener-Weinberg approach [8] to the Coleman-Weinberg potential, i.e. we renormalize at scale $\mu$ at which the running self-coupling vanishes, $\lambda(\mu) = 0$. $\Phi_{PQ}$ always couples to some fermions. For definiteness we focus on the KSVZ model which introduces a heavy quark $Q_L, Q_R$ (charged under the chiral $U(1)_{PQ}$) and Yukawa coupling $g\Phi_{PQ}\bar{Q}_L Q_R + (h.c.)$. Contribution of $Q$ to the Coleman-Weinberg potential $V_{CW}$ tends to destabilize it. Potential $V_{CW}$ may be bounded from below if $\Phi_{PQ}$ couples also to some bosons. For simplicity we will consider $N_\phi$ copies of a scalar singlet $\phi$, of bare mass $m_\phi$ and the following coupling to $\Phi_{PQ}$ and non-minimal coupling to gravity:

$$\lambda_{mix}\, \phi^2 \Phi_{PQ}^\dagger \Phi_{PQ} + \frac{1}{2}\xi_\phi \mathcal{R}\phi^2\,, \tag{3}$$

where $\mathcal{R}$ denotes the Ricci scalar. Neglecting effects caused by non-zero temperature and non-zero curvature of the space-time, the CW potential reads

$$V_{CW} = \frac{1}{64\pi^2}\left\{ N_\phi \mathcal{M}_\phi^4\left[\log\left(\frac{\mathcal{M}_\phi^2}{\mu^2}\right) - \frac{3}{2}\right] - N_Q \mathcal{M}_Q^4\left[\log\left(\frac{\mathcal{M}_Q^2}{\mu^2}\right) - \frac{3}{2}\right]\right\}\,, \tag{4}$$

where $\mathcal{M}_\phi^2 = m_\phi^2 + \lambda_{mix}S^2$ and $\mathcal{M}_Q^2 = \frac{g^2}{2}S^2$. However, it occurs that in many cases the curvature effects may change quite substantially characteristics of the CW potential. In de Sitter approximation for the inflationary space-time, $\mathcal{R} = 12H_I^2$. $V_{CW}$ generalizes to [9]:

$$V_{\text{inf}} \propto \frac{1}{64\pi^2}\left\{\mathcal{M}_\phi^4\left[\log\frac{|\mathcal{M}_\phi|^2}{\mu^2} - \frac{3}{2}\right] - \mathcal{M}_Q^4\left[\log\frac{|\mathcal{M}_Q|^2}{\mu^2} - \frac{3}{2}\right] - \frac{1}{15}H_I^4\log\frac{|\mathcal{M}_\phi|^2}{\mu^2} + \frac{38}{15}H_I^4\log\frac{|\mathcal{M}_Q|^2}{\mu^2}\right\}\,, \tag{5}$$

where now $\mathcal{M}_\phi^2 = m_\phi^2 + (\xi_\phi - \frac{1}{6})12H_I^2 + \lambda_{mix}S^2$ and $\mathcal{M}_Q^2 = H_I^2 + \frac{g^2}{2}S^2$. Since $V$ is somewhat complicated, in the remaining we will focus on a particular, simplified case. First, we choose that the bosonic and fermionic contributions are similar: $N_\phi = N_Q = 4 \cdot 3$ and $\lambda_{mix} = 1/2g^2(1+\epsilon)$, where $0 < \epsilon \ll 1$. This corresponds to a quasi "SUSY" limit. Second, we assume that $\phi$s do not feature the inflationary stochastic fluctuations, $\mathcal{M}_\phi^2(S=0) > \frac{1}{4}H_I^2$ [10]. Finally, we assume that the non-minimal coupling $\xi_{PQ}\mathcal{R}S^2$ is negligible.

An interesting feature of $V$ emerges after it is expanded in small $S$, $V = C_2 \cdot S^2 + C_4 \cdot S^4 + \dots$ One obtains $C_4 \propto \left(\lambda_{mix}^2 \log\frac{\mathcal{M}_\phi^2(S=0)}{\mu^2} - (\frac{g^2}{2})^2 \log\frac{H_I^2}{\mu^2}\right)$. Hence, $C_4 < 0$ if $H_I$ is not too small, which implies that $V$ has a (second) minimum at a somewhat large scale. This situation is illustrated in fig. 1 for exemplary choices of couplings and dimensionful parameters. One can see for somewhat intermediate $H_I$, the $U(1)_{PQ}$ can be restored, while for yet larger $H_I$ there

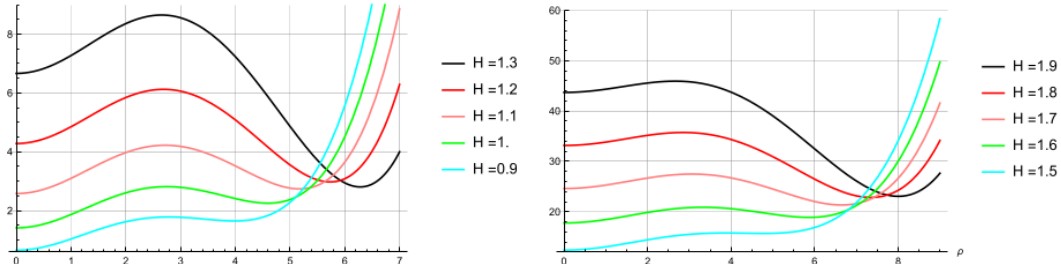

Figure 1: $V/\mu^4$ as function of $S/\mu$ for $\xi_\phi = 1/6, \lambda_{mix} = 0.2, g^2 = 0.18$; $m_\phi/\mu = 0.5(1.)$ in the left (right) plot.

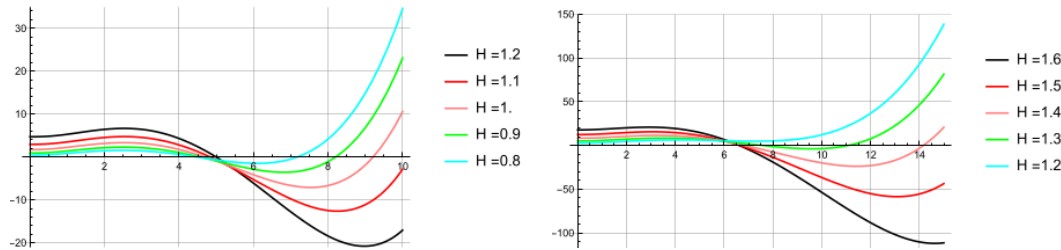

Figure 2: $V/\mu^4$ as function of $S/\mu$ for $\xi_\phi = 1/6, \lambda_{mix} = 0.2$; $g^2 = 0.19(0.196), m_\phi/\mu = 0.2(1.)$ in the left (right) plot.

indeed emerges a global minimum at larger scale. Moreover, in fig. 2 it is illustrated that the scale $S$ of the second (global) minimum grows as the "SUSY" limit is approached. Notice, that around the global minimum $V'' \ll O(H_I^2)$, so in this scenario the stochastic dynamics determines $S_i$ provided long enough inflation. The asymptomatic Fokker-Planck probability distribution for the radial mode $S$ for $\epsilon \ll 1$ has a small variance around the minimum.

More quantitatively, the following estimate holds for $S_i$ if $H_I$ is not too small (in the sense outlined above) and the large scale minimum occurs: $S_i^2 \sim O(1) \cdot 1/\epsilon \cdot H_I^2/g^2$. On the other hand, if $H_I$ is too small one simply obtains $S_i^2 \sim O(1) \cdot H_I^2/g^2$. Such "SUSY" enhancement of $S_i$ could be desirable in various non-thermal mechanisms for DM generation. For example, if stochastic fluctuations $\delta\theta$ survive until $V_a$ is generated and the misalignment mechanism constitutes total DM relic, then the enhancement with $\epsilon = 10^{-3}$ relaxes bounds on the couplings due to constraints from CDM isocurvature to order $\lambda_{mix}, g^2 \lesssim 10^{-5}$. Same constraints but without the enhancement imply a significantly stronger bound, $\lambda_{mix}, g^2 \lesssim 10^{-8}$. In the next section we argue that $\delta\theta$ can survive until $V_a$ emerges.

# 3 Post-Inflationary dynamics of $\Phi_{PQ}$ in the radiative potential

For $\delta\theta$ fluctuations to survive until $V_a$ is generated, it suffices that oscillations of the *radial* mode $S$ survive until a global minimum emerges at some $S \neq 0$ in the post-inflationary $S$ potential, which we will denote by $V$. Indeed, in such a case, the memory of the angular direction of the radial oscillations survive, hence $\delta\theta$ survive. Whether particles $\phi, Q$ constitute components of a thermal bath, depends, in particular, on the underlying model of reheating. Hence, thermal corrections to $V$ might be absent; for simplicity we assume the latter for the remaining discussion.[1] For some time after the end of inflation, due to the Hubble friction, the field $S$ has a constant value $S_i$. It starts to oscillate (in the post-inflationary analogue of (5))

---

[1] In some cases substantial thermal corrections, if present, may change some features of a model. However, this depends on details of a given model. Detailed analysis of such cases is beyond the scope of this work.

when the value of the Hubble parameter decreases to $H_i$ (equal about one third of the effective mass of the radial mode which depends on $S_i$ so also on $H_I$) approximately given by

$$H_i^2 \approx H_I^2 \frac{3\lambda_{mix}}{16\pi^2}\left[(1-4\xi_\phi) - \frac{2m_\phi^2}{9H_I^2}\right]\ln\left(\frac{(3-12\xi_\phi)H_I^2 - m_\phi^2}{2\epsilon\mu^2}\right) \ll H_I^2. \qquad (6)$$

Thus, typically in our model $S$ field starts to oscillate when the reheating process is completed, if the reheating efficiency is big enough.

We will denote the critical Hubble parameter, below which the curvature effects do not suffice to restore $U(1)_{PQ}$, by $H_c$. To show that axion fluctuations can survive, below we simply list a couple of concrete "benchmarks" for which $H_c \gg H_i$; for $\xi_\phi = \frac{1}{6}$ and $\xi_\phi = \frac{3}{16}$ respectively:

$$\{\epsilon = 10^{-3}, \ \lambda_{mix} = 1.4\cdot 10^{-5}, \ \mu = 1.3\cdot 10^{12} \text{ GeV}, \ H_I = 1.5\cdot 10^{12} \text{ GeV}, \ m = 7.7\cdot 10^{11} \text{ GeV}\},$$
$$\{\epsilon = 10^{-2}, \ \lambda_{mix} = 1.3\cdot 10^{-6}, \ \mu = 4\cdot 10^9 \text{ GeV}, \ H_I = 2\cdot 10^{12} \text{ GeV}, \ m = 2\cdot 10^9 \text{ GeV}\}. \qquad (7)$$

The examples were chosen such that the CDM isocurvature constraints are fulfilled, the ALP condensate saturates CDM relic density and $H_I \gg 10^{10}$ GeV. Some features of our model are as in "unbroken $U(1)_{PQ}$" models: $H_I$ may be large, minimum of $V(\Phi)$ during and after inflation is different than the final one related to the axion decay constant, PQ field evolves during and after inflation. On the other hand, non-vanishing $\Phi$ during inflation as well as characteristics of axion DM and its isocurvature perturbations are typical for "broken $U(1)_{PQ}$" case.

## 4 Conclusions

The traditional description of $\Phi_{PQ}$ dynamics in the "PQ unbroken scenario" ($f_a \ll H_I$) is often too simplified. For the quartic $\Phi_{PQ}$ potential this is the case if $\lambda \ll O(0.01)$. Instead of staying at zero, the radial component $S$ of $\Phi_{PQ}$ acquires large initial value $S_i$ (almost homogeneous) in the observable Universe, despite the presence of temperature corrections.

In various non-thermal mechanisms axion/ALP DM generation, very large $S_i$ is considered, which requires extremely small $\lambda$. In such cases radiative corrections could be important and it is therefore relevant to investigate their role. To this aim, we studied $\Phi_{PQ}$ dynamics in the CW potential in the inflationary and post-inflationary Universe. We used as an example the quasi "SUSY" limit in which the bosonic and fermionic contributions to CW are of very similar magnitude. We found that curvature effects result in significant enhancement of $S_i$, if $H_I$ is just not too small, i.e. at least of order of the scale of the CW potential.

The otherwise desirable enhancement of $S_i$ also suppresses the inflationary fluctuations of the axion. If these fluctuations are not washed out until the axion potential emerges, they constitute the CDM isocuvature mode. Since the latter is strongly constrained, the enhancement of $S_i$ weakens the corresponding bounds on the couplings in the CW potential. Finally, we discussed the (ALP) case with negligible thermal corrections to the PQ potential in the post-inflationary Universe, to argue that ALP fluctuations may survive in natural and phenomenologically interesting parameter regions.

## Acknowledgements

**Funding information** Work supported by National Science Centre, Poland, grant DEC-2018/31/B/ST2/02283.

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
