# Peer review of "Axion-like particle as Cold Dark Matter via the misalignment mechanism with PQ symmetry unbroken during inflation"

_SciPost Physics Proceedings, doi:SciPost Phys. Proc. 12, 038 (2023)_

## Round 1 · Referee Report · Anonymous (Referee 1) · 2022-11-10

Strengths

  1. This work studies the feeble-$\lambda$ case of the PQ potential, and argues that here the curvature corrections can be crucial. Especially, in the quasi ``SUSY'' limit, a second (global) VEV may appear, leading to suppressed isocurvature perturbations.

  2. The topic of this manuscript is well motivated. And the idea to suppress stochastic fluctuations by achieving an earlier second (global) VEV from curvature effect is very interesting.

Weaknesses

  1. The statement in Section.3 is a little confusing for me. Does $S$ obtain a non-zero VEV during the inflation, or after the inflation? If this new proposal is more like a PQ broken'' instead ofPQ unbroken'' scenario, the author may mention it explicitly.

  2. If I understand it correctly, the first equality in Eq. (6) is only true during the inflation, since the curvature effects become time-dependent in the post-inflationary era, right?

  3. The temperature correction is always neglected, while potentially it may lead to interesting effects.

Report

This submission definitely meets the criteria of SciPost Physics Proceedings, and it should be published here after several minor improvements are made.

Requested changes

  1. See the weakness 1 above.

  2. The abstract mentions that "We find that many such models predict unacceptable isocurvature perturbations", but this is not explicitly discussed in the main text. I suggest the author remove or improve this sentence.

  3. "in order to obtain bounded from below $V_{CW}$" sounds confusing, probably change it to "to make sure $V_{CW}$ is bounded from below" or so?

---

## Round 2 · Referee Report · Anonymous (Referee 1) · 2022-11-22

Report

The authors have answered the questions in the previous report. I really appreciate the texts added to the end of Sec.3, and would happily recommend the publication of this manuscript in SciPost Physics Proceedings, just after the very naive question below.

I still hope to understand Eq. (6) better. In a (post-inflationary) radiation-dominated Universe, the curvature correction should be a function of temperature $T$, being independent of inflation Hubble parameter $H_I$. So, could the authors explain explicitly why $H_I$ appears in the first equality of Eq. (6), but $T$ does not. Is $H_I$ simply from $S_i$, and $T$-terms are negligible, or I am missing something?

BTW, at the end of footnote 1, it should be "the scope of this work".

---

## Round 2 · Author Response

We would like to thank the referee for his/her report and constructive suggestions
for further improving the presentation. We have addressed all of the referee's points, in particular by making
modifications to the text, which are marked as red in the new version (2nd). A description of the modifications together with answers to the Referee's questions or comments, are listed below.

---

## Round 2 · List of Changes

"1. The statement in Section.3 is a little confusing for me. Does SS obtain a non-zero VEV during the inflation, or after the inflation? If this new proposal is more like a “PQ broken'' instead of “PQ unbroken'' scenario, the author may mention it explicitly."

S obtains VEV during inflation.

  • Concerning the first issue, the text was further improved by modifying the sentence which contains eq. (6) and the former, on page 4.
  • An extra comment on the second issue was added at the end of the current verion. It starts at the very end of the main text on page 4.

"2. If I understand it correctly, the first equality in Eq. (6) is only true during the inflation, since the curvature effects become time-dependent in the post-inflationary era, right?"

The first equality in Eq. (6) is for the post-inflationary era. It already accounts for the fact that the curvature effects to the potential become time-dependent after inflation. Ihe improvements to the text above eq. (6) described above already address clarification of this issue.

"3. The temperature correction is always neglected, while potentially it may lead to interesting effects."

  • A corresponding comment was added in the footnote on page 4.

"2. The abstract mentions that "We find that many such models predict unacceptable isocurvature perturbations", but this is not explicitly discussed in the main text. I suggest the author remove or improve this sentence."

  • We followed the Referee's suggestion and removed this sentence.

"3. "in order to obtain bounded from below VCW sounds confusing, probably change it to "to make sure VCWVCW is bounded from below" or so?"

  • The sentence above eq. (3) was modified accordingly.

---

## Round 3 · Referee Report · Anonymous (Referee 1) · 2022-11-23

Report

Thanks for the clear answer from the authors. Now I fully recommend it for publication in SciPost Physics Proceedings.

---

## Round 3 · Author Response

We would like to thank the referee for his/her report. We have addressed the remaining referee's point. Correspondingly, we made a modification to the text, now marked in blue. If the referee recognizes the modification as redundant, we could go back to the previous version (2nd).

We hope that our answers now dispels the referee's doubts.

We list the modifications and answers below.

---

## Round 3 · List of Changes

"The authors have answered the questions in the previous report. I really appreciate the texts added to the end of Sec.3, and would happily recommend the publication of this manuscript in SciPost Physics Proceedings, just after the very naive question below.

I still hope to understand Eq. (6) better. In a (post-inflationary) radiation-dominated Universe, the curvature correction should be a function of temperature T, being independent of inflation Hubble parameter HI. So, could the authors explain explicitly why HI appears in the first equality of Eq. (6), but T does not. Is HI simply from Si, and T-terms are negligible, or I am missing something?"

Yes, the dependence in the first equality in eq. (6), on H_I is from S_i .

On one hand, in this work the thermal corrections are assumed absent (according to the already introduced footnote on page 4).

On the other hand, the curvature effects in the post-inflationary universe depend on the temperature only indirectly - they depend on the (changing) Hubble parameter. Then, yes, the corresponding temperature dependence of the first equality in eq. (6) would be very small - hence the approximation sign in the equality.

  • The sentence containing eq.(6) was modified accordingly.

"BTW, at the end of footnote 1, it should be "the scope of this work"."

  • The typo is now corrected.

---

## Editorial Decision

published